# From Coffee Waste to Active Ingredient for Cosmetic Applications

**DOI:** 10.3390/ijms24108516

**Published:** 2023-05-10

**Authors:** Giovanna Grigolon, Kathrin Nowak, Stéphane Poigny, Jane Hubert, Alexis Kotland, Laura Waldschütz, Franziska Wandrey

**Affiliations:** 1Mibelle Group Biochemistry, Mibelle AG, 5033 Buchs, Switzerland; giovanna.grigolon@mibellegroup.com (G.G.); kathrin.nowak@mibellegroup.com (K.N.); stephane.poigny@mibellegroup.com (S.P.); 2NatExplore SAS, 51140 Prouilly, France; jane.hubert@nat-explore.com (J.H.); alexis.kotland@nat-explore.com (A.K.); 3NATECO2-Hopfenveredlung St. Johann GmbH, 85283 Wolnzach, Germany; contact@nateco2.de

**Keywords:** sustainability, coffee silverskin, kahweol, cafestol, chemical profiling, diterpenoid esters, eco-friendly extraction, skin-barrier, hydration

## Abstract

Coffee silverskin (CS) is the thin epidermis covering and protecting the coffee bean and it represents the main by-product of the coffee roasting process. CS has recently gained attention due to its high content in bioactive molecules and the growing interest in valuable reutilization of waste products. Drawing inspiration from its biological function, here its potential in cosmetic applications was investigated. CS was recovered from one of the largest coffee roasters located in Switzerland and processed through supercritical CO_2_ extraction, thereby generating coffee silverskin extract. Chemical profiling of this extract revealed the presence of potent molecules, among which cafestol and kahweol fatty acid esters, as well as acylglycerols, β-sitosterol and caffeine. The CS extract was then dissolved in organic shea butter, yielding the cosmetic active ingredient SLVR’Coffee™. In vitro gene expression studies performed on keratinocytes showed an upregulation of genes involved in oxidative stress responses and skin-barrier functionality upon treatment with the coffee silverskin extract. In vivo, our active protected the skin against Sodium Lauryl Sulfate (SLS)-induced irritation and accelerated its recovery. Furthermore, this active extract improved measured as well as perceived skin hydration in female volunteers, making it an innovative, bioinspired ingredient that comforts the skin and benefits the environment.

## 1. Introduction

Current environmental challenges pose the need for a re-evaluation of production processes and techniques across industries. It has become essential to evaluate the quality of a product not only by its usefulness and efficacy, but also by the sustainability of its production process and its ecotoxicity. The valorization of waste-products fully meets this need and fosters the migration to a more circular economy, thereby reducing pollution. Food processing by-products are a valuable source of precious metabolites that can be given a second life through industrial biotechnology. Agro-industrial coffee residues include coffee pulp, husks, hulls, defective beans and coffee silverskin (CS) [1,2]. CS is a thin, closely fitting tegument enveloping each of the two coffee beans found in the coffee cherry [3]. CS is further encapsulated in a looser layer, called the parchment, surrounded by the pulp of the coffee fruit [4]. Prior to being roasted, the coffee beans contain approximately 11% moisture, which is then converted into steam through the heat of the roasting process. The steam causes the bean to expand and the silverskin to split from the coffee bean and fall off, thereby constituting the biggest by-product of the roasting procedure. In total, 9.9 billion kg of coffee are produced yearly worldwide, leading to 200–400 million kg of CS generated [5]. The disposal of CS as solid waste in landfills, as a fertilizer or through combustion may have negative effects on the environment and thus requires proper management [6]. Interestingly, previous characterization studies reported CS to contain health-promoting bioactive molecules, such as phenols, flavonoids, phytosterols and alkaloids [7]. Furthermore, CS was described to have antioxidant, antimicrobial, antifungal and prebiotic properties [3,8,9], whilst being at the same time skin compatible and safe [10]. The richness of its composition, together with the rising consumer demand for more natural and sustainable ingredients that improve skin appearance make CS an ideal candidate for cosmetic formulations. In particular, CS showed potential to protect the skin from chemical, environmental and mechanical stress, as well as from microbial infection, by intervening on the outermost layer of the epidermis, the *Stratum corneum* [1]. This is a multilayer of keratinocytes embedded in a matrix of lipids of both sebaceous and epidermal origin. Keratinocyte-produced lipids fill intercellular spaces and are a mixture of cholesterol, free fatty acids and ceramides, while sebaceous lipids are mainly nonpolar, such as triglycerides, squalene and wax esters [11]. The *Stratum corneum* protects against external aggressors, retains water and transports nutrients and its lipids are the main mediators of the permeability of the skin barrier. In fact, reduced barrier function is often characterized by an abnormal disposition of these lipids, compared to healthy skin [12]. For the present study, we aimed at developing a biomimicry concept inspired by the protective role of CS, also described as the epidermis of the coffee bean. We asked whether our cosmetic active ingredient SLVR’Coffee^TM^ (coffee silverskin extract dissolved in organic Union for Ethical Biotrade (UEBT)-certified shea butter, from here on termed “coffee silverskin active” or “CSA”), obtained through supercritical CO_2_ extraction of CS waste, could exert a protective function on the human skin, just as the coffee silverskin envelops and protects the delicate coffee bean during its growth.

## 2. Results

### 2.1. Coffee Silverskin Sourcing and Supercritical CO_2_ Extraction

CS was sourced from one of the largest coffee roasters located in Switzerland. The flakes from the coffee roasting process were first pressed into pellets and then subjected to supercritical CO_2_ extraction. In total, 1435 kg of starting material resulted in 33 kg of coffee silverskin extract and 1355 kg of extraction residue. This is equivalent to an extraction yield of 2.3%. While the extraction residue was composted for biogas production, the precious coffee silverskin extract underwent composition analysis and in vitro testing. The liposoluble extract was then dissolved in organic UEBT-certified shea butter at low temperature (≤60 °C) to avoid degradation of bioactive molecules. In this way, the coffee silverskin extract was further developed into an active ingredient that can be used for cosmetic formulations.

### 2.2. Chemical Profiling of the Coffee Silverskin Extract

The coffee silverskin extract was obtained by supercritical CO_2_ extraction and thus exhibited a low polarity. Therefore, a relatively nonpolar biphasic solvent system composed of *n*-heptane and methanol (1/1, *v*/*v*) was used to produce by centrifugal partition chromatography (CPC) a series of fractions of simplified chemical composition. A total of 11 final fractions was obtained, containing CS metabolites organized in a decreasing order of polarity (Figure 1). From the ^13^C NMR spectra of the 11 fractions, the chemical shift values and signal intensities were collected and a bucketing strategy resulted in a table with 11 columns (one per fraction) and 258 rows (one per chemical shift bin containing a ^13^C NMR peak detected in at least one fraction). This table was submitted to hierarchical clustering analysis (HCA) on the rows. In this way, statistical correlations between ^13^C NMR peak fingerprints belonging to a single structure and thus evolving similarly in successive fractions were easily visualized as “chemical shift clusters” in the heatmap (Figure 2). By confronting these clusters to our predictive NMR database of natural molecules, followed by manual interpretation of 2D NMR data for validation or further structure elucidation, several significant constituents of the coffee silverskin extract were identified. As a result, we found that the first fractions F01-F02, representing together ≈53% of the extract mass, were mainly composed of di- and triacylglycerols, as well as β-sitosterol. In the fractions F03-F06, which represented ≈24% of the extract mass, a range of diterpene fatty acid esters derived from cafestol and kahweol were detected as largely major constituents. Among these kaurene-type diterpenes, cafestol linoleate and kahweol linoleate were unambiguously identified by NMR. Other minor derivatives were detected, but neither NMR nor LC/MS analyses helped to 100% confirm the nature of their fatty acid chains, probably due to poor ionization capacity of these metabolites in our analytical conditions. Based on literature data, we can suspect the prevalence of palmitate, linoleate and stearate fatty acid chains [13,14,15,16]. Fractions F07 and F08 which represented together ≈7.5% of the extract mass, were mainly composed of sterols, mainly stigmastane-3,6-dione, and free fatty acids. Finally, in the most polar fractions F09-F11 recovered over the extrusion step and representing ≈15% of the extract mass, caffeine was identified as the largely major constituent, followed by other diterpene fatty acid esters among which there were 19-Norkaura-1,4-(18)-diene-18-carboxaldehyde 16,17-dihydroxy-3-oxo derivatives.

### 2.3. Gene Expression Analysis on Keratinocytes

Coffee silverskin extract tested at 0.04% stimulated the gene expression in keratinocytes of different markers related to oxidative stress response and skin barrier function. The upregulation of HMOX1 (encoding heme oxygenase 1) by 137% indicated an improved antioxidant response. Furthermore, the expression of the genes CALML5 (calmodulin-like protein 5), FLG (filaggrin) and SPRR1A (small proline-rich protein 1A) involved in the cornification process were increased by 124%, 150% and 125%, respectively (Table 1).

### 2.4. Coffee Silverskin Active (CSA) Reduces Skin Sensitivity

Skin reactivity was investigated after 28 days of application of 2% coffee silverskin active (CSA). The evaluation of the stinging test with lactic acid demonstrated that the skin sensitivity was significantly reduced by 92.1% compared to initial conditions. Moreover, skin reactivity was significantly improved compared to placebo (Figure 3).

### 2.5. Coffee Silverskin Active Rehydrates Dry Legs

Treatment of dry legs with 2% CSA for 14 and 28 days led to an improvement of skin hydration compared to placebo treatment. In comparison to the initial conditions, skin hydration was significantly increased by 26.8% and 33.1% after treatment with 2% CSA for 2 and 4 weeks, respectively (Figure 4). The positive outcome was also evident from the self-questionnaire where 95% of volunteers described their skin to be less scaly (Table 2). In total, 100% of volunteers reported that their skin was more hydrated and less dry after treatment with 2% CSA. Together, the results highlight that CSA has a moisturizing and rehydrating effect.

### 2.6. Coffee Silverskin Active Protects against SLS Irritation

Application of 2% CSA for 7 days before irritation with sodium lauryl sulfate (SLS) had a protective effect on the skin. After the removal of the SLS patch, increases in redness and microcirculation, which are indicative of irritation, were reduced in the CSA-treated condition by 9% and 36.9%, respectively, compared to untreated conditions. Reductions in microcirculation were significantly improved compared to placebo. Furthermore, the increase in transepidermal water loss (TEWL), indicative of skin barrier disruption, was significantly prevented (26.6%) compared to untreated control (Figure 5).

### 2.7. Coffee Silverskin Active Induces Faster Skin Regeneration after SLS Irritation

Treatment with CSA after irritation with SLS had a regenerative effect on the skin. The recovery time of the skin parameters redness, microcirculation and TEWL was significantly reduced compared to untreated conditions (19.4%, 7.6% and 11%, respectively). In addition, the recovery time improvement in redness, microcirculation and TEWL was significant compared to placebo treatment (Figure 6). The positive outcome was also evident from the questionnaire where 95.2% of volunteers described that the treatment with CSA hydrated their skin and made their skin smoother. In total, 90.5% of the volunteers reported that the treatment with CSA reduced skin dryness after irritation (Table 3).

## 3. Discussion

In the present study, an innovative cosmetic ingredient obtained from the main by-product of the coffee roasting process, coffee silverskin, was developed and characterized. The coffee silverskin used for this process was sourced from a local coffee roastery, allowing for a short production chain and for the minimization of environmental impact.

To upcycle this waste material, coffee silverskin was converted into pellets and a non-polar extract was generated through supercritical CO_2_ extraction. The application of the generally recognized as safe solvent CO_2_ in combination with high pressure presents several advantages over conventional procedures. Supercritical CO_2_ extraction was demonstrated to yield chemically stable products of superior quality, rich in bioactive molecules and free of solvent traces [17]. Moreover, CO_2_ was recycled sustainably during this process and the generated waste could be used for biogas production. This environmentally friendly technique, often associated with the concept of “green chemistry” is especially attractive for its applications in the cosmetic industry. The preparation of lipophilic plant extracts through supercritical CO_2_ extraction is particularly convenient because a complete lipophilic spectrum of the plant constituents can be obtained without dilution, stress of high temperatures or decomposition of the plant ingredients. Moreover, it allows for a better penetration of the solvent into the plant material, given the lower surface tension, lower viscosity and better diffusion achievable [17]. Hence, despite the low yield of the extraction process, a precious and rich material could be obtained through this extraction technique. The chemical profile of the CS extract was analyzed using a ^13^C NMR-based dereplication procedure, leading to the identification of several potent metabolites. Among those, di- and triacylglycerols were found. The lipolytic hydrolysis of epidermal di- and tryacylglycerols generates fatty acids that participate in the synthesis of acylceramides. These contribute to the formation of the cornified lipid envelope, the interface between corneocyte proteins and the extracellular matrix in the *stratum corneum*. Hence, the presence of di- and triacylglycerols is very important for maintaining permeability barrier homeostasis [18]. β-sitosterol is a plant phytosterol which has been demonstrated to exhibit anti-inflammatory, anti-cancer and immune-modulating properties [19] and was highly present in the coffee silverskin extract. An in vivo study performed in mice could show that this phytosterol improved clinical symptoms of atopic dermatitis and reduced scratching behavior following histamine-treatment [19]. Furthermore, β-sitosterol enhanced the expression of hyaluronic acid synthases in fibroblasts and that of genes encoding skin barrier functional proteins in keratinocytes [20]. A range of diterpene fatty acid esters derived from cafestol and kahweol was also enriched in the CS extract. In particular, cafestol linoleate and kahweol linoleate were identified through NMR. We hypothesize that microbiome-secreted skin enzymes, such as esterases and lipases found in the epidermis, can convert these precursor molecules into the diterpenes kahweol, cafestol and free fatty acids [21,22,23,24,25]. Cafestol and kahweol are natural diterpenes mostly known for their detrimental effect on serum lipids levels when taken orally [26,27]. However, the same molecules displayed extensive anti-inflammatory, anti-cancer, anti-diabetic and anti-osteoclastogenetic activity in other studies [28,29,30,31,32]. Notably, kahweol promoted skin-moisturizing activities by upregulating STAT1 in HaCaT cells [33]. Cafestol and kahweol, as well as bengalensol and caffeine, also highlighted by high resolution LC/MS, have been described as potent activators of bitter taste receptors, which are not only expressed in gustatory cells but also in skin cells [34,35]. In human keratinocytes, bitter taste receptors showed a protective role against skin aging by regulating cellular senescence and wound healing [36]. Sterols (mainly stigmastane-3,6-dione) and free fatty acids were also identified in the CS extract, as well as caffeine and other diterpene fatty acid esters, among which there were 19-Norkaura-1,4-(18)-diene-18-carboxaldehyde 16,17-dihydroxy-3-oxo derivatives. The beneficial effects of caffeine in skin applications have been extensively described. This alkaloid protects skin from oxidative stress-induced senescence through activating autophagy [37], counteracts the effect of UV radiation, slows down photoaging and increases the microcirculation of blood in the skin [38].

Many studies have linked the occurrence of sensitive skin to epidermal barrier function disruption [39,40,41]. A weak epidermal barrier is caused by a derangement of the intercellular lipids surrounding the corneocytes in the *stratum corneum*. This facilitates the penetration of irritants and allergens, exposes nerve endings and increases TEWL, resulting in the perception of skin discomfort [42]. In this study, the CS extract reduced skin sensitivity and reactivity during a lactic acid stinging test, probably through upregulation of the expression of genes involved in the cornification process. Among those, filaggrin (FLG) functions in the transition of keratinocytes to corneocytes and might effectively control keratinocyte differentiation and controlled cell death. Although its structural and functional relationship with accessory molecules such as actins is still not fully understood, its role in skin barrier formation has been established [43,44]. Chlorogenic acids found in green coffee beans have been found to upregulate FLG expression [45], but further studies are needed to better understand the interaction between the active molecules contained in the CS extract and the pathways involved in filaggrin synthesis. Filaggrin is also considered part of the natural moisturizing factor in the skin, since its degradation products play an important role in the water holding capacity of the *stratum corneum* [46]. Consistently, CS extract showed a significant effect in enhancing perceived as well as measured skin hydration on legs. Moreover, immediate protection from SLS stress and a significantly faster recovery could be achieved through the application of CSA compared to placebo control. A significant reduction in redness, TEWL and microcirculation impairment demonstrated the potent effect of the molecular cocktail contained in the CSA.

## 4. Materials and Methods

### 4.1. Coffee Silverskin Supercritical CO_2_ Extraction

Supercritical CO_2_ extraction of coffee silverskin was carried out using large-scale equipment. The extraction system consisted of stainless-steel vessels with a total volume of 1500 L. Temperature and pressure were measured inside the vessels and independently modified. A specific extraction pressure of 250–300 bar, a temperature between 40 and 60 °C and a CO_2_/starting material ratio of 20 to 50 were applied. The bulk density of the milled starting material ranged between 550–600 kg/m^3^ and the vessel was loaded accordingly with coffee silverskin. The technical process of CO_2_ extraction is depicted in Figure 7. The extraction process is initiated by pressurization and temperature increase, thereby modifying the density of liquid CO_2_ to obtain supercritical CO_2_, which flows through the extraction vessel and pulls along soluble compounds (coffee silverskin extract). A subsequent pressure reduction changes the solubility characteristics of CO_2_, and the extract is divided, falling into separation vessels (so-called separators). The gradual regulation of the extraction parameters enables the optional generation of different fractions. For this application, only separator 1 was used to generate a total extract without fractionation. To recover the used CO_2_, the condenser liquefies it and reintroduces it again into the enclosed cycle.

### 4.2. Centrifugal Partition Chromatography (CPC)

Methanol, *n*-heptane, and acetone were purchased from Carlo Erba Reactifs SDS (Val de Reuil, France). Aqueous solutions were prepared with ultrapure water. Deuterated dimethyl sulfoxide (DMSO-*d6*) was purchased from Sigma-Aldrich (Saint-Quentin, France).

CPC fractionation was performed on a FCPE300^®^ rotor (Rousselet Robatel Kromaton, Annonnay, France) made of seven partition disks containing in total 231 twin cells (≈1 mL per cell, 303 mL total column capacity) and connected to a Knauer Preparative 1800 V7115 pump (Berlin, Germany). A biphasic solvent system was prepared by mixing *n*-heptane and methanol in the proportions 1/1 (*v*/*v*) in a separating funnel (total volume ≈ 3 L). The CPC column was filled with the lower phase of the biphasic solvent system (used as the stationary phase) at 500 rpm and 50 mL/min, and then equilibrated in the ascending mode with the upper phase (used as the mobile phase) at 1300 rpm and 20 mL/min. In parallel, the sample solution was prepared by directly solubilizing 864 mg of the CS extract in 10 mL of lower phase and 3 mL of upper phase. The sample was loaded into the column using an 18 mL injection loop, and the mobile phase was pumped in the ascending mode for 35 min at 20 mL/min (elution step). Then the column was extruded by switching the mode from ascending to descending for 10 min, still at 20 mL/min. Fractions of 20 mL were collected over the whole experiment using a Pharmacia Superfrac collector (Uppsala, Sweden). The fractionation was assessed by thin layer chromatography (TLC) using a CAMAG^®^ automatic sampler ATS4, a CAMAG^®^ automatic developing chamber ADC2, and a CAMAG^®^ TLC visualizer 2. Fractions were deposited on pre-coated silica gel 60 F254 Merck plates were used with the migration solvent system *n*-heptane/acetone (5/1, *v*/*v*), visualized under UV light at 254 nm and 366 nm and revealed by spraying the dried plates with 50% H_2_SO_4_ and vanillin, followed by heating. The fractions were combined based on their TLC profile similarities, resulting in a final series of 11 fractions (Figure 1).

### 4.3. NMR Analyses and Metabolite Identification

An aliquot of each fraction from F01 to F11 (up to 15 mg when possible) was dissolved in 600 µL of DMSO-*d6*. NMR analyses were performed at 298 K on a Bruker Avance AVIII-600 spectrometer (Karlsruhe, Germany) equipped with a TXI cryoprobe optimized for 1H detection and with cooled 1H and ^13^C coils and preamplifiers. ^13^C NMR spectra were acquired at 150.91 MHz. An udeft pulse program was used with an acquisition time of 0.3599 s. For each sample, 512 scans were coadded to obtain a satisfactory signal-to-noise ratio. The spectral width was 238.9070 ppm, and the receiver gain was set to the highest possible value. The spectra were manually phased and baseline corrected using the TOPSPIN 4.0.5 software (Bruker) and calibrated on the central resonance of DMSO-d6 (δ 39.80 ppm). The absolute intensities of ^13^C NMR signals were automatically collected and the collected peaks in all fractions were subsequently sorted in a final table using a bucketing script written in Python language. The principle of this script was to divide the ^13^C spectral window from 0 to 238.9070 ppm into regular chemical shift intervals (Δδ = 0.2 ppm) and to associate the absolute intensity of the collected peaks to the corresponding bin. The resulting table was imported into the Permut Matrix version 1.9.3 software (LIRMM, Montpellier, France) for hierarchical clustering analysis (HCA) on peak intensity values. The classification was performed on the rows only, i.e., on the chemical shift buckets. The Euclidian distance was used to measure the proximity between samples, and the Ward’s method was applied to agglomerate the data. The resulting ^13^C NMR chemical shift clusters were visualized as dendrograms on a 2D map. The higher the intensity of ^13^C NMR peaks, the brighter the yellow color on the map (Figure 2). For metabolite identification, each ^13^C NMR chemical shift cluster obtained from HCA was manually submitted to the structure search engine of a ^13^C NMR chemical shift database (ACD/NMR Workbook Suite 2012 software, ACD/Laboratories, Toronto, ON, Canada) comprising the predicted chemical shifts and structures of natural molecules (≈8300 records in March 2023). Additionally, 2D NMR experiments (HSQC, HMBC, and COSY) were recorded for all fractions to confirm or further elucidate the chemical structures proposed by the database at the end of the dereplication process. The major metabolites identified in each fraction are presented in Figure 2.

### 4.4. High Resolution Liquid Chromatography Mass Spectrometry

All CPC fractions were also analyzed by high resolution LC/MS with an Acquity UPLC H-Class (Waters, Manchester, UK) system coupled to a Synapt G2-Si from Waters equipped with an electrospray (ESI) ion source. The chromatographic column was an Uptisphere C-18 ODB 150 × 4.6 mm, 5 µm from Interchim, maintained at 35 °C. The mobile phase gradient started with 100% solvent A (MilliQ water + 0.1% formic acid) and 0% solvent B (acetonitrile + 0.1% formic acid), then increased to 65% solvent B in 10 min, to 100% solvent B from 10 to 13 min, remained at 100% solvent B for 7 min, and then recycled back to 100% solvent A from 20 to 21 min and remained at 100% solvent A until 26 min. The flow rate was set at 0.7 mL/min and the sample injection volume was 1 µL. MS acquisition was performed in the positive ion mode within the scan range 50 < *m*/*z* < 2000. The capillary voltage was set at 2 kV, the desolvation temperature at 450 °C, the desolvation gas flow at 950 L/h, the source temperature at 120 °C, the cone voltage at 40 V and the cone gas flow at 50 L/h. Data were processed using the MassLynx V4.2 software from Waters (Manchester, UK). LC/MS data are summarized in Appendix A.

### 4.5. Keratinocyte Culture and Treatment

Normal human epidermal keratinocytes (NHEK, used at the 3rd passage) were seeded in 24-well plates and cultured for 24 h (37 °C, 5% CO_2_) in culture medium (Keratinocyte-SFM optimized for the assay, supplemented with epidermal growth factor and pituitary extract) and in assay medium (Keratinocyte-SFM optimized for the assay) for a further 24 h. The medium was then replaced by assay medium containing or not (control) the CS extract or the solvent control (DMSO tested at 0.04%) and the cells were incubated for 24 h. All experimental conditions were performed in *n* = 3. At the end of incubation, the cells were washed in phosphate buffered saline (PBS) solution and immediately frozen at −80 °C.

### 4.6. Differential Expression Analysis

The expression of markers was analyzed using RT-qPCR method on total RNA extracted from the cell monolayers of each experimental condition (before RNA extraction, the replicates of the same experimental condition were pooled). The analysis of transcripts was performed in *n* = 2 using a PCR array targeting 64 genes selected for their importance in skin physiology.

### 4.7. RNA Extraction and Reverse Transcription

Total RNA was extracted from each sample using TriPure Isolation Reagent^®^ according to the supplier’s instructions. The amount and quality of RNA were evaluated using capillary electrophoresis (Bioanalyzer 2100, Agilent technologies, Santa Clara, CA, USA). The complementary DNA (cDNA) was synthetized by reverse transcription of total RNA in the presence of oligo(dT) and « Transcriptor Reverse Transcriptase » (Roche). The cDNA quantities were then adjusted before the PCR step.

### 4.8. Quantitative PCR

The PCR (Polymerase Chain Reaction) were performed using the “LightCycler^®^” system from Roche Molecular System Inc. (Pleasanton, CA, USA) according to the supplier’s instructions.

The reaction mix (10 μL final) was prepared as follows:-2.5 μL of cDNA,-primers (forward and reverse),-reagent mix (Ozyme) containing taq DNA polymerase, SYBR Green I and MgCl_2_.

### 4.9. Data Management of Quantitative PCR

Raw data were analyzed using Microsoft Excel^®^ software V2302 from Microsoft Corporation (Redmond, WA, USA). The incorporation of fluorescence in amplified DNA was continuously measured during the PCR cycles. This resulted in a “fluorescence intensity” versus “PCR cycle” plot, allowing the evaluation of a relative expression (RE) value for each marker. The value selected for RE calculations is the “output point” (Ct) of the fluorescence curve. For a considered marker, the highest is the cycle number; the lowest is the mRNA quantity. The RE value was calculated with the formula: (1/2 number of cycles) × 106. The PCR array included 3 reference genes (RPS28, GAPDH and ACTB). These housekeeping genes were used for data normalization since their expression is constitutive and theoretically stable. Consequently, the level of expression of the target markers was compared to the mean expression level of these 3 markers for all test conditions.

### 4.10. In Vivo Skin Protection and Regeneration Test

The clinical study was performed as a single center, blinded, randomized and controlled study in healthy subjects. The evaluation of the subjects was compared against the baseline evaluation and between products. This study was performed according to the Declaration of Helsinki principles and subsequent amendments. In this study, 24 female subjects with phototype II and III with stinging positive skin (skin reactive to lactic acid application) on the face and with an age between 18 and 65 years (mean: 46.8 years) were included in the study. There were 3 dropouts and no withdrawals and therefore the skin acceptability and compatibility and the efficacy of the test products were assessed on 21 subjects. Subjects with cutaneous marks on the experimental areas, which could interfere with the assessment of skin reactions, with allergy or reactivity to products of the same category as the tested one and subjects that were exposed to the sun for a prolonged time in the month preceding the study were excluded. Moreover, subjects who had been receiving topical treatment using corticoids on the application site in the eight days prior to starting the study, or treatment with Vitamin A or its derivatives in the 3 months before the beginning of the study or UVA or UVB treatment in the month preceding the study were also excluded, as well as pregnant or breast-feeding women, subjects forecasting intensive sun or UVA exposure during the test period, subjects practicing intensive sport whose temporary interruption creates difficulties, subjects forecasting vaccination during the test period or the three weeks before its start and subjects carrying visible eczematoid reactions, scars or pigmentary sequelae of previous tests in the experimental area. Protection and regeneration studies were performed on the forearm area. The stinging test evaluation was performed on the nasolabial fold on the face. All the evaluations were performed in a fully controlled room and after an initial acclimatization process of at least 30 min in a fully controlled and acclimatized room (controlled temperature: T = 21 ± 2 °C; controlled relative humidity: RH = 55 ± 10%). To avoid circadian changes, 2 evaluation periods were defined (morning: 9h00 m–13h30 m; afternoon: 13h30 m−18h30 m). Each subject had chosen, on the first day, the best hour to be evaluated. The following evaluations were performed over the same period of the day.

For all tests, a gel containing 2% CSA or the same gel composition without active (placebo) was applied twice daily.

The instruments used for measuring the tested parameters were, for skin microcirculation: Periflux PF5000 (Perimed, Jakobsberg, Sweden), for skin redness: a* parameter, Chromameter^®^ CR-400 (Minolta, Osaka, Japan), for TEWL: Tewameter^®^ TM300 (Courage + Khazaka, Cologne, Germany).

For the protection test, measurement of test parameters was performed on day 1, followed by the first test product application. From day 1 to day 7, the test products were applied twice daily on one forearm 2% coffee silverskin active and the placebo on the other forearm, according to a randomization plan. On day 7 or 8, the test parameters were measured, and the subjects were exposed to 2% SLS for 24 h under an occlusive patch. On day 8, test parameters were measured again after SLS patch removal. For the regeneration test, on day 1, test parameters were measured, and the subjects were exposed to 2% SLS for 24 h under occlusion. From day 2 to day 24, the products were applied twice daily, and measurements of the test parameters were taken every 2–3 days during this time window. Recovery time was defined by the time needed for the respective parameter to reach baseline value + 15%.

### 4.11. Lactic Acid Stinging Test

On day 1, the base value stinging test evaluation was performed with 10% lactic acid solution applied on the nasolabial fold, followed by a self-assessment of the stinging sensation immediately after application, as well as after 2 min, 5 min and 8 min. The stinging score is calculated as a mean value of the last three assessment times. Afterwards, the first test product application was performed on the face. From day 1 to day 28 the test products were applied twice daily, on one face half 2% coffee silverskin active and the placebo on the other face half, according to a randomization plan. On day 29, the stinging test evaluation was performed with 10% lactic acid solution applied on the nasolabial fold as described above. Then the subjects were asked to fill a product efficacy questionnaire.

### 4.12. In Vivo Hydration Measurement on Dry Legs

This study has been conducted pursuant to the Declaration of Helsinki (1864), with the amendments of Tokyo (1975), Venice (1983), Hong Kong (1989) and Seoul (2008). Subjects with a history of any form of skin cancer, melanoma, lupus, psoriasis, connective tissue disease, diabetes or any disease that would increase risk associated with study participation were excluded, as well as subjects having relevant cutaneous marks in the experimental area or receiving aesthetic treatments on this area finishing less than 2 months before the start of the study. Individuals undergoing a medical treatment possibly interfering with the instrumental measurements, pregnant or lactating females (within the 6-months period preceding the start of the study) as well as subjects showing allergy or reactivity to any of the components of the test product or those forecasting a relevant change of routine during the study period were also excluded. In total, 20 out of the 22 recruited volunteers, aged from 31 to 65 years (mean 43.7), completed the study. Before the start of the treatment, subjects attended the test facilities for the baseline measurements. Upon arrival, volunteers were allowed to acclimatize to environmental conditions (21 ± 2 °C, 45 ± 10% humidity) for 15 min. Then, hydration skin levels were quantified in both legs through Corneometer CM 825, and an initial photograph of the area of interest was taken. The same parameters were recorded in the area of interest after 2 or 4 weeks of topical application. Data recorded from every individual at the different time points were normalized versus baseline values (Day 0) from the same individual and statistically analyzed for each treatment separately. In addition, participant’s subjective perception of the product efficacy was assessed with an individual questionnaire. The studied parameter was recorded from the legs, with 5 technical measurements performed on the upper half (thighs) and other 5 technical measurements performed on the lower region (from the knee to the ankle), using Corneometer CM 825.

### 4.13. Statistical Analysis

Unpaired *t*-tests and two-way ANOVA statistical analysis were performed using GraphPad Prism 8.0.1.

## Figures and Tables

**Figure 1 ijms-24-08516-f001:**
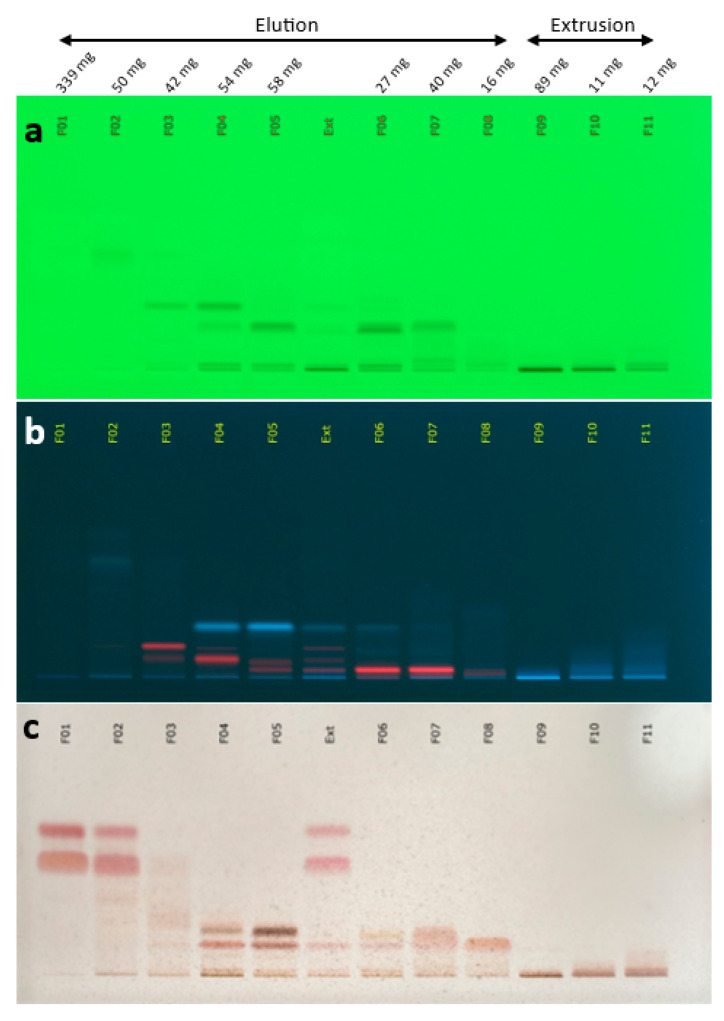
Thin layer chromatography (TLC) profile of the 11 final centrifugal partition chromatography (CPC) fractions (F01-F11) produced from the coffee silverskin CO_2_ extract (Ext). (**a**) 254 nm; (**b**) 366 nm; (**c**) visible after vanillin/H_2_SO_4_ reagent spraying and heating.

**Figure 2 ijms-24-08516-f002:**
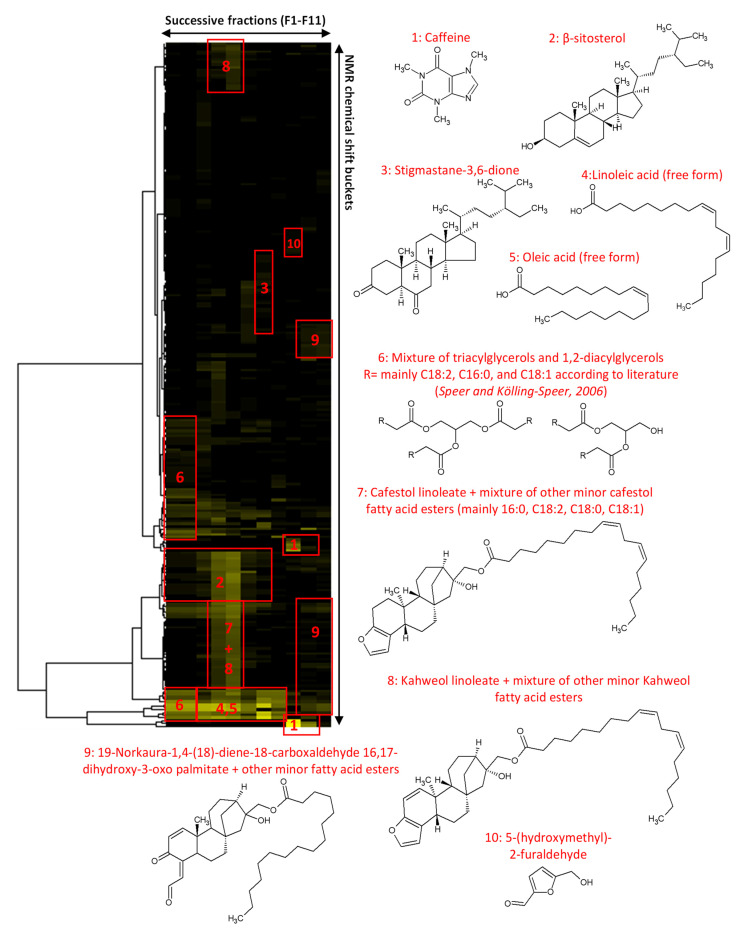
Hierarchical clustering analysis (HCA) heat map of ^13^C NMR signals with the identified compounds.

**Figure 3 ijms-24-08516-f003:**
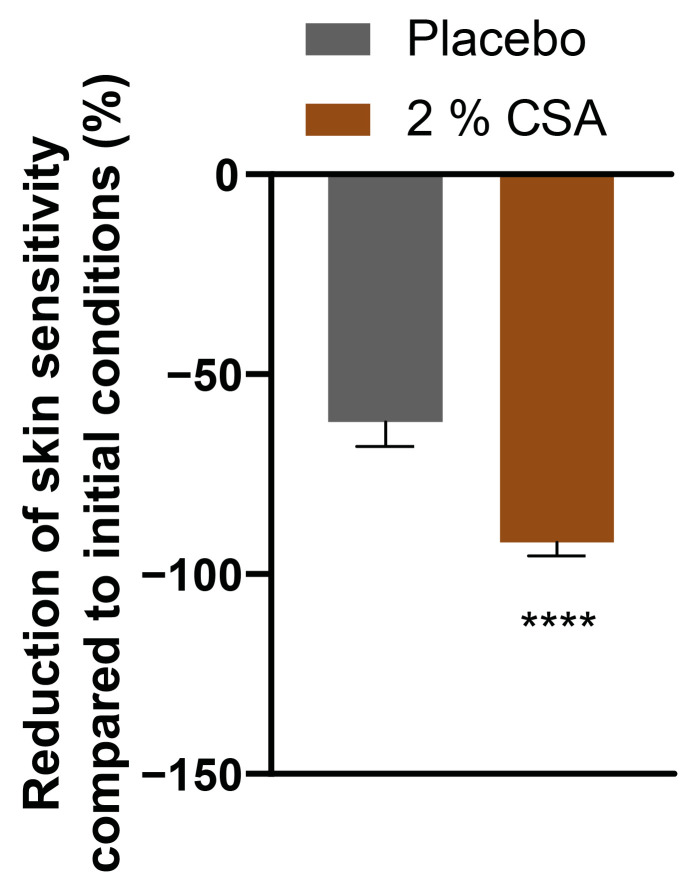
Reduction of skin sensitivity in the nasolabial fold after 28 days of test material application. Unpaired *t*-test. **** *p* ≤ 0.0001 compared to placebo treatment.

**Figure 4 ijms-24-08516-f004:**
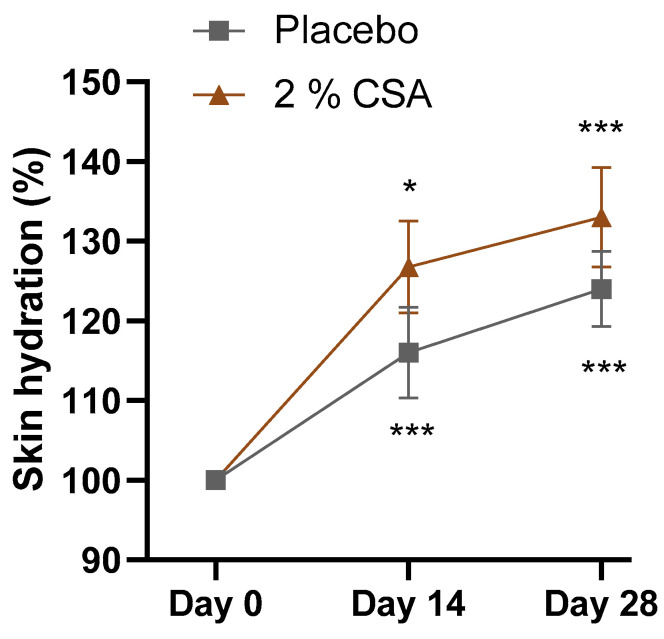
Graphical representation of the hydration content (%), before (day 0) and after 2 (day 14) or 4 (day 28) weeks of treatment with placebo or 2% CSA in 21 volunteers. Values obtained at D14 or D28 were normalized to the corresponding levels at D0. Mean and standard error of the mean (SEM) are shown. Two-way ANOVA statistical analysis was applied for significance between treatments and initial condition. * *p* ≤ 0.05, *** *p* ≤ 0.001.

**Figure 5 ijms-24-08516-f005:**
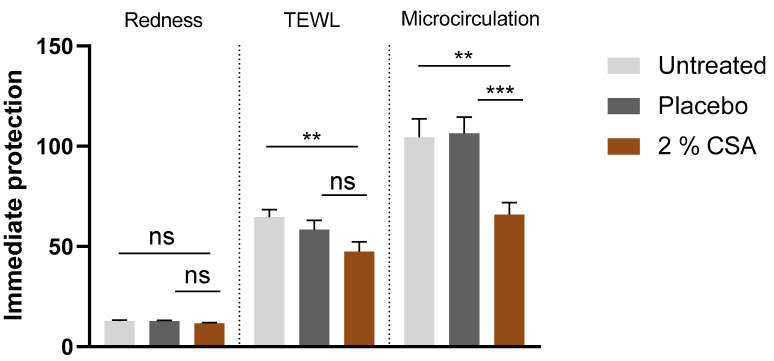
Preventive protection against SLS irritation. Measured units for redness: AU; TEWL: g/h/m^2^; microcirculation: AU. Unpaired *t*-test. *** *p* ≤ 0.001, ** *p* ≤ 0.01, ns *p* > 0.05.

**Figure 6 ijms-24-08516-f006:**
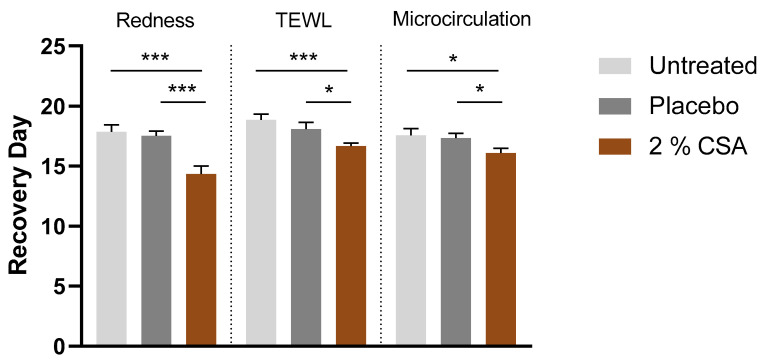
Regenerative effect of coffee silverskin active after SLS irritation. Measured units for redness: AU; TEWL: g/h/m^2^; microcirculation: AU. Unpaired *t*-test. *** *p* ≤ 0.001, * *p* ≤ 0.05.

**Figure 7 ijms-24-08516-f007:**
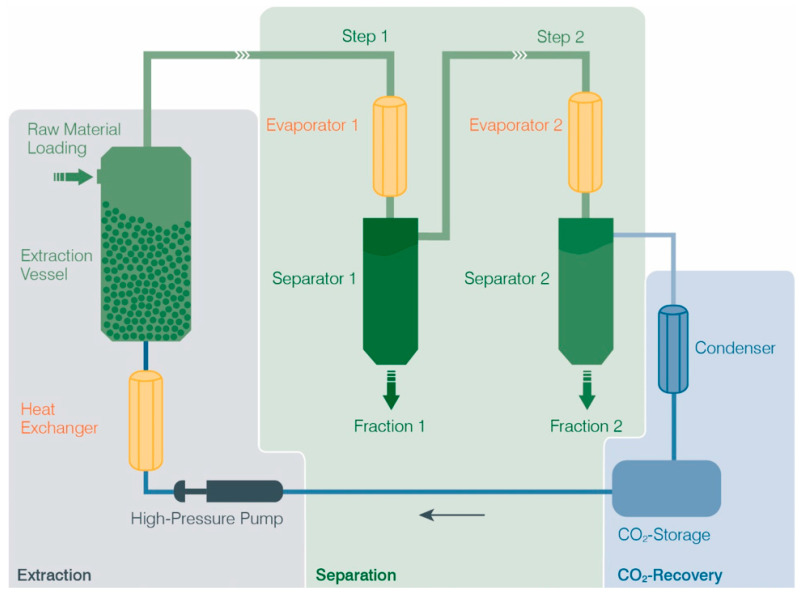
Supercritical CO_2_ extraction process. ©Hopfenveredlung St. Johann GmbH (NATECO_2_), Germany.

**Table 1 ijms-24-08516-t001:** Gene expression changes upon treatment of keratinocytes with coffee silverskin extract.

Gene Name	Gene Expression Compared to Control (%)
HMOX1	137%
CALML5	124%
FLG	150%
SPRR1A	125%

**Table 2 ijms-24-08516-t002:** Summary of answers obtained in the self-assessment questionnaire for the study on dry legs.

	% Satisfaction Placebo	% Satisfaction CSA
My skin seems smoothed	81	86
My skin is less red	81	90
My skin looks younger	67	71
My skin is more hydrated	81	100
My skin is less dry	86	100
My skin seems firmer	71	86
My skin seems more elastic (flexible)	76	86
My skin seems more radiant	67	71
My skin is less scaly	90	95

**Table 3 ijms-24-08516-t003:** Summary of answers obtained in the self-assessment questionnaire for the study with SLS irritation.

	% Satisfaction Placebo	% Satisfaction CSA
The treatment hydrates the skin	85.7	95.2
The treatment makes my skin smoother	81.0	95.2
The treatment reduces skin redness	81.0	85.7
The treatment prepares my skin for irritation	71.4	76.2
The treatment calms my skin after irritation	85.7	90.5
The treatment helps my skin to recover faster from irritation	85.7	90.5
The treatment reduces the itchiness after irritation	85.7	90.5
The treatment reduces the stinging after irritation	85.7	90.5
The treatment homogenizes my skin tone after irritation	71.4	81.0
The treatment reduces skin dryness after irritation	81.0	90.5

## Data Availability

Not applicable.

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
