# Peer review of "From Coffee Waste to Active Ingredient for Cosmetic Applications"

_ijms, 2023, doi:10.3390/ijms24108516_

Round 1

Reviewer 1 Report

The current study deals with the extraction of coffee silverskin (a coffee waste) and the application of the extract in the cosmetic industry. The scope of the study is clearly stated. Moreover, the experimental part is well organized, and the manuscript is well written. Statistical analysis of the results has been conducted and the results are well presented and discussed with those of other authors in the literature. Here are some comments:

-   -       Abstract: Please include SLS in full, as well (not only in abbreviated form).

-   -       Abstract (lns 23-24): Please add the word “compound” or other next to “active”. Although, it is explained later in the manuscript it looks like something is missing when someone reads the abstract.

-        -  Introduction (lns 45-46): Please be consistent, use either kg or kilograms.

-        -  Why was the percentage of 2% CSA chosen?

-        -  Paragraph 4.5: I believe the authors would like to write -80 °C.

-        -  Please add the statistical analysis tools used in a different paragraph in the M&M section.

Author Response

Response to Reviewer 1 Comments

Thank you for your positive comments to our manuscript and for your suggestions, which we implemented. Please see the point-by-point response to each of your comments.

Comments and Suggestions for Authors

The current study deals with the extraction of coffee silverskin (a coffee waste) and the application of the extract in the cosmetic industry. The scope of the study is clearly stated. Moreover, the experimental part is well organized, and the manuscript is well written. Statistical analysis of the results has been conducted and the results are well presented and discussed with those of other authors in the literature. Here are some comments:

-   Abstract: Please include SLS in full, as well (not only in abbreviated form). Done

-   Abstract (lns 23-24): Please add the word “compound” or other next to “active”. Although, it is explained later in the manuscript it looks like something is missing when someone reads the abstract. Added: “extract”

 -  Introduction (lns 45-46): Please be consistent, use either kg or kilograms. Done and checked throughout the manuscript.

 -  Why was the percentage of 2% CSA chosen? We optimized the concentration of the CSA in a cosmetic formulation to ensure clinical efficacy as well as cost-effectiveness whilst the formulation properties (stability, color etc) are not negatively affected. CSA has additionally been evaluated in two previous clinical studies at use levels of 1-2% and showed good skin compatibility and acceptability. 2% CSA was therefore chosen as the most effective concentration.

 -  Paragraph 4.5: I believe the authors would like to write -80 °C. Corrected

-  Please add the statistical analysis tools used in a different paragraph in the M&M section. Done

Reviewer 2 Report

The work ijms-2381917 is a clear example of bioprospecting, in which coffee silver skin, an abundant residue of the coffee agroindustry, was used to obtain bioactive ingredients for commercial cosmetic use (https://mibellebiochemistry.com/slvrcoffeetm). Overall, the manuscript is very well written, its methodologies are clear, and the results are well discussed, which was the product of a clear multidisciplinary work. In addition, the references are relevant.

There are few recommendations that I will give, but most revolve around better use of units of measurement and reporting of data. Here are a few:

- I would recommend following the same system of units throughout the entire text. Given the nature of the journal, I recommend implementing the guideline from the National Institute of Standards and Technology (NIST). For this purpose, it should be noted that each numerical value must be accompanied by its respective unit, and, in addition, this unit must be separated by a space from the numerical value (i.e., 10 °C, 20 °C and 30 °C).

- I am not in favor of separate Results and Discussions sections. I believe that this extends the work and makes us lose the opportunity to go side by side analyzing the results that are presented. I recommend that the authors merge both sections.

L263: The correct unit is “L”. This must be corrected throughout the manuscript.

L297: The correct unit is “min”. This must be corrected throughout the manuscript.

Table 4: I recommend providing it as supplementary material.

L364: The correct unit is “h”. This must be corrected throughout the manuscript.

L406: in vivo is not capitalized, even when it starts a sentence or a title. Apply this throughout the manuscript.

Author Response

Response to Reviewer 2 Comments

Thank you for the positive feedback on our manuscript. We appreciate the time you dedicated to it and value your comments and suggestions. Please see the point-by-point response to each of your comments.

Comments and Suggestions for Authors

The work ijms-2381917 is a clear example of bioprospecting, in which coffee silver skin, an abundant residue of the coffee agroindustry, was used to obtain bioactive ingredients for commercial cosmetic use (https://mibellebiochemistry.com/slvrcoffeetm). Overall, the manuscript is very well written, its methodologies are clear, and the results are well discussed, which was the product of a clear multidisciplinary work. In addition, the references are relevant.

There are few recommendations that I will give, but most revolve around better use of units of measurement and reporting of data. Here are a few:

- I would recommend following the same system of units throughout the entire text. Given the nature of the journal, I recommend implementing the guideline from the National Institute of Standards and Technology (NIST). For this purpose, it should be noted that each numerical value must be accompanied by its respective unit, and, in addition, this unit must be separated by a space from the numerical value (i.e., 10 °C, 20 °C and 30 °C). Done and checked throughout the manuscript.

- I am not in favor of separate Results and Discussions sections. I believe that this extends the work and makes us lose the opportunity to go side by side analyzing the results that are presented. I recommend that the authors merge both sections. We appreciate this suggestion; however, we are structuring the manuscript according to the journal guidelines. Moreover, given the multidisciplinary nature of this work, we think that separate results and discussion sections could help the reader distinguish the different steps required for the preparation of the active ingredient and its testing.

- L263: The correct unit is “L”. This must be corrected throughout the manuscript. Done and checked throughout the manuscript.

- L297: The correct unit is “min”. This must be corrected throughout the manuscript. Done and checked throughout the manuscript.

- Table 4: I recommend providing it as supplementary material. It was moved to supplementary material.

- L364: The correct unit is “h”. This must be corrected throughout the manuscript. Done and checked throughout the manuscript.

- L406: in vivo is not capitalized, even when it starts a sentence or a title. Apply this throughout the manuscript. We removed the capitalization and changed the font to normal (before: italic), according to the authors guidelines.